# The Interplay of Inter- and Intramolecular Hydrogen Bonding in Ether Alcohols Related to n-Octanol

**DOI:** 10.3390/molecules30112456

**Published:** 2025-06-04

**Authors:** Markus M. Hoffmann, Troy N. Smith, Gerd Buntkowsky

**Affiliations:** 1Department of Chemistry and Biochemistry, State University of New York Brockport, Brockport, NY 14420, USA; tsmit31@brockport.edu; 2Institute of Physical Chemistry, Technical University Darmstadt, Peter-Grünberg-Straße 8, D-64287 Darmstadt, Germany

**Keywords:** molecular dynamics simulations, hydrogen bonding, radial distribution functions, density, viscosity, self-diffusion coefficients, alcohols, ethers

## Abstract

n-Octanol and related ether alcohols are studied via molecular dynamics (MD) simulations using the two classical all-atom force fields OPLS-AA and CHARMM. The ether alcohols studied possess one ether functionality separated by varying n carbon atoms from the hydroxy group to elucidate how the positioning of the ether functionality affects intra- and intermolecular hydrogen bonding and, in turn, the physical properties of the studied alcohols. Important general trends observed from simulations with both force fields include the following: Intramolecular hydrogen bonding is majorly present in 3-butoxypropanol and 4-propoxybutanol (n = 3 and 4) while being only marginally present for 5-ethoxypentanol and 6-methoxyhexanol (n = 5 and 6) and absent in 1-hexyloxymethanol and 2-pentyloxyethanol (n = 1 and 2). The intramolecular hydrogen bonds formed by 3-butoxypropanol and 4-propoxybutanol are among the most stable ones of all present hydrogen bonds. Intermolecular hydrogen bonding is stronger between hydroxy groups (OH-OH) than between hydroxy and ether groups (OH-OE). An increased temperature causes a reduction in intermolecular OH-OH and OH-OE hydrogen bonding but a slight increase in intramolecular hydrogen bonding. A reduction in end-to-end distances at a higher temperature is also observed for all studied alcohols, which is likely a reflection of increased dihedral bond rotations. Hydrogen bonding extends mostly between just two molecules while hydrogen bonding networks are rare but do exist, involving, in some instances, up to 30 hydrogen bonds. Regardless of force field and temperature, the obtained radial distribution functions (RDFs) mostly show the same features at same distances that only vary in their intensity. 1-hexyloxymethanol forms a very specific and stable intermolecular double OH-OE hydrogen-bonded dimer. Similar double-hydrogen-bonded dimers can be found for the ether alcohols but are only significantly present for 2-pentyloxyethanol. Overall, the main difference between OPLS-AA and CHARMM is their quantitative prediction of the present hydrogen bonding speciation largely due to the stiffer dihedral potentials in OPLS-AA compared to the CHARMM force field. The simulations indicate that (a) the variations in densities are correlated to the reduced packing efficiency caused by intramolecular hydrogen bonding, (b) self-diffusion correlates with the stability of the intermolecular hydrogen bonds, and (c) the presence of hydrogen-bonded networks, although small in numbers, affect the viscosity.

## 1. Introduction

In a recent experimental study [1], new physical property data of density, viscosity, and self-diffusion were reported for n-octanol and related ether alcohols. This molecular dynamics (MD) study seeks to obtain a molecular-level understanding of these alcohols to be able to explain several interesting structure–property trends observed in this prior experimental study. Particular attention is paid to a careful analysis of the hydrogen bonding behavior in these molecules. As can be seen from the molecular structures shown in Table 1, the ether alcohols possess one hydroxyl functional group and one ether functional group. Therefore, these molecules can engage only in the following three different types of hydrogen bonding: intermolecular hydrogen bonding between two hydroxyl groups (OH-OH), intermolecular hydrogen bonding between hydroxyl and ether functional groups (OH-OE), and intramolecular hydrogen bonding between hydroxyl and ether functional groups. This MD study seeks to understand how the interplay between these three types of hydrogen bonding scenarios is dependent on the position of the ether functional group in the ether alcohols in Table 1. Further connections can then be drawn to the resulting physical properties of these molecules allowing for a molecular-level explanation of structure–property relationships.

The insights obtained in this study are potentially of interest to many other areas of chemistry research. Hydrogen bonding is prevalent in many chemical and biological systems, including proteins, carbohydrates, and nucleic acids, and when present, may play essential roles in how molecules assemble and react [2]. Some of the present major research areas involving hydrogen bonding interactions include the formation of supramolecular structures [3,4] such as protein-integrated hydrogen-bonded frameworks [5] and deep eutectic solvents (DESs), which are composed of molecules that are hydrogen bond donors and acceptors [6]. Our research area of interest is to establish a molecular-level understanding of polyethylene glycols (H-[O-CH_2_-CH_2_]_n_-OH), PEGs) as a chemical solvent [7,8,9,10]. Liquid PEGs with a molar weight less than 1000 g·mol^−1^ have proven to be an effective solvent in chemical synthesis [11,12,13], as these can dissolve a wide variety of chemicals [14], even some mineral salts [15]. PEG is also an attractive solvent because of its environmentally benign characteristics, including its non-toxicity, low vapor pressure, and biodegradability [16,17]. Furthermore, PEGs are widely available, as they are produced industrially mostly as an ingredient in health and personal care products [16]. Compared to the ether alcohols in this study, the hydrogen bonding interactions in PEG are much more complex because the molecular structure of each ethylene glycol oligomer in the polydisperse PEG solvent includes two hydroxy groups and a varying number of ether functionalities depending on the size of the ethylene glycol oligomer. Interestingly, a recent simulation of PEG200, where the number 200 reflects the average molar weight, indicated a high propensity for tetraethylene glycol and, to a smaller extent, also for triethylene glycol to form intramolecular hydrogen bonds [9]. This present MD study was, in part, motivated to shed more light on the origin of this peculiar propensity.

It should be noted that research is ongoing to develop further the fundamental understanding of hydrogen bonding. The latest 2011 recommendation by IUPAC defines a hydrogen bond as follows [18]: “The hydrogen bond is an attractive interaction between a hydrogen atom from a molecule or a molecular fragment X–H in which X is more electronegative than H, and an atom or a group of atoms in the same or a different molecule, in which there is evidence of bond formation”. The forces in hydrogen bonding are said to “include those of an electrostatic origin, those arising from charge transfer between the donor and acceptor leading to partial covalent bond formation between H and Y, and those originating from dispersion”. Van der Lubbe and Guerra further specify a total of seven energy contributions to hydrogen bonding: electrostatic interactions, charge transfer interactions, π-interaction assistance, cooperative effects, Pauli repulsion, dispersion interactions, and secondary interactions [19]. Renati and Madl criticize that such a description of the forces is based on a corpuscular view that, with the general quantum–theoretical approach, falls short in deriving the underlying dynamics [20]. They claim that the application of Quantum Field Theory–Quantum Electro-Dynamics leads to a more accurate description of coherence domains necessary to describe hydrogen-bonded networks in systems such as water. No such quantum mechanical details are considered in this study. Instead, the MD simulations presented here are based on classical force fields where hydrogen bond numbers are determined based on simple bond distance and bond angle criteria. This is justified, as semi-quantitative results are sufficient for the goal of this benchmark study to discern trends in hydrogen bonding across the ether alcohols in Table 1. A more refined and quantitative hydrogen bonding analysis would require a combination of complementary computational and experimental approaches, as illustrated by Pocheć et al. for n-octanol [21], and of ab initio metadynamics calculations, as proposed by Biswas and Wong [22], which are beyond the scope of this investigation. Classical force fields have been frequently used for many systems, including biomolecules, where the most widely employed force field families are [6] AMBER [23,24], CHARMM [25,26], GROMOS [27], and OPLS-AA [28]. Two force fields were chosen for repeat simulations in this study to test how far the observed trends are independent of force field choice. The two force fields picked are OPLS-AA and CHARMM. As shown in the Computational Methods section, both force fields are quite similar and have been referred to as Type 1 force fields [29]. The OPLS-AA force field was picked because it has been found to reproduce the closest experimental densities, viscosities, and self-diffusion coefficients in a recent study on PEG200 [9]. The CHARMM force field was chosen because one of its main differences in parametrization concerns the dihedral potential functions. These have been shown to be important in improving the accuracy of the dynamic properties self-diffusion and viscosity for PEG200 [9].

n-Octanol is included in this MD study for two reasons. Firstly, there are prior MD simulations on n-octanol available [21,30,31,32] to allow for a comparison of the simulated results in contrast to the ether alcohols, which, to the best of our knowledge, have not been explored in theoretical studies. Secondly, since 1-octanol is of a very similar molar weight to the ether alcohols in Table 1 and its molecular structure is devoid of any ether functionality, it serves as a reference molecule to assess the effect of the ether functionality on the system’s structure and dynamics. In addition, there are noteworthy practical applications of n-octanol as an amphiphile molecule to model membranes [32,33] and to measure hydro/lipophilicity of solutes of interest [34].

## 2. Computational Methods

### 2.1. Simulation Details

The general simulation protocol was carried out with the freely available software package GROMACS 2020.4 [35,36] and consisted of the following steps: Using the “insert molecules” module, 1000 molecules were inserted randomly into a virtual box at least 20% larger in size than the volume calculated from the substance’s density. Atoms that were in too close contact with each other were allowed to shift position by 0.01 nm increments using the “steepest descent” algorithm to remove these high potential energy configurations. The box was then allowed to equilibrate its volume at a constant number of particles, constant pressure of one 1 bar, and constant temperature (constant NPT), which took less than 1 ns. The constant NPT simulation continued for a total of 10 ns. The average volume of the equilibrated simulation box was determined and used for the final step of the simulation protocol data, which was a simulation at a constant number of particles, constant volume, and constant temperature (constant NVT) for a total of 100 ns. The constant temperature in this simulation protocol was either set at 298 K or 358 K, as these represent the lowest and highest temperatures of a prior experimental study [1]. As can be seen in Equation (1), the potential function, *U_total_*, for describing the atom-to-atom interactions consist of the same contributions for both force fields: bond vibrations, *U_bond_*, angular vibrations, *U_angles_*, bond torsions, *U_torsion_*, and the nonbonded Lenard Jones (LJ) dispersion and Coulomb interactions, *U_nonbond_*.(1)$$\begin{array}{c} U_{total}=U_{bonds}+U_{angles}+U_{torsion}+U_{nonbond}\\ U_{bonds}=\sum_{bonds}\frac{1}{2}k_{b}{\left(b-b_{o}\right)}^{2}\\ U_{angles,OPLS-AA}=\sum_{angles}\frac{1}{2}k_{\theta }{\left(\theta -\theta_{o}\right)}^{2}\\ U_{angles,CHARMM}=\sum_{angles}\frac{1}{2}k_{\theta }{\left(\theta -\theta_{o}\right)}^{2}+\frac{1}{2}k_{UB}{\left(S-S_{o}\right)}^{2}\\ U_{torsion,OPLS-AA}=\sum_{\begin{array}{c} torsions \end{array}}\sum_{n=0}^{5}C_{n}\cos^{n}\left(\phi -180{}^{\circ}\right)\\ U_{torsion,CHARMM}=\sum_{\begin{array}{c} \;proper\\ torsions \end{array}}k_{\phi }\left(1+\cos\left(n\phi -\delta \right)\right)\\ U_{nonbonded}=\sum_{i<j}4f_{LJ}\epsilon_{ij}\left[{\left(\frac{\sigma_{ij}}{r_{ij}}\right)}^{12}-{\left(\frac{\sigma_{ij}}{r_{ij}}\right)}^{6}\right]+\sum_{i<j}\frac{f_{qq}}{4\pi \epsilon_{o}}\frac{q_{i}q_{j}}{r_{ij}} \end{array}$$

The meaning of the various parameters in Equation (1) is as follows: $k_{b}$, $k_{\theta }$, *k_UB_*, and $k_{\theta }$ are, respectively, the bond, angle, Urey–Bradley and dihedral angle force constants; *b*, θ, *S*, and ϕ are, respectively, the bond length, bond angle, Urey–Bradley 1,3-distance, and dihedral angle, where the subscript zero represents the respective equilibrium value; *C_n_* are the torsional energy barrier coefficients of the Ryckaert–Bellemans potential for proper dihedrals; *q* is the partial Coulomb charge on a particular atom; *σ* and *ε* are, respectively, the Lennard–Jones (LJ) contact distance and well depth; and *f_qq_* and *f_LJ_* are fudge factors for the Coulomb and LJ interactions, respectively. The fudge factors only apply for the OPLS-AA force field, i.e., they are set to 1 for the CHARMM force field. For the OPLS-AA force field, they are set to 0.5 for 1-4 atom pairs, to zero for 1-2 and 1-3 atom pairs, and 1 for all other atom pairs. The CHARMM force field uses as combination rule arithmetic means for the *σ* parameter and geometric means for the *ε*, while the OPLS-AA force field uses geometric means for both parameters.

As can be seen in Equation (1), the main formal differences between the two force fields lie in the treatment of the angles and torsion potentials. For the torsion potentials, the differences manifest themselves in dihedral angle positions that more easily transition for the CHARMM force field, as exemplarily shown in Appendix A.

Other simulation details were similar for both force fields. The simulation time step was 2 fs, and velocities were generated for each atom using a Maxwell–Boltzmann distribution. The motion of all bonds involving hydrogen atoms were restrained using the LINCS algorithm with a fourth-order matrix expansion [37]. Periodic boundary conditions were applied using the Verlet cutoff scheme [38] with a buffer tolerance of 0.005 kJ mol^−1^ ps^−1^ and the Smooth Particle-Mesh Ewald (PME) scheme [39,40] to account for the long-range electrostatic interactions. The cutoff distances and grid spacing differed between the two force fields, namely 1.4 nm and 0.168, respectively, for OPLS and 1.2 nm and 0.144 nm for the CHARMM force field. A drift of the center of mass of the system was checked for and, if detected, corrected every 20 fs. With respect to temperature and pressure coupling (1 bar), the Bussi–Donadio–Parrinello velocity-rescaling thermostat [41], with a time constant of 1.0 ps, and the Parrinello–Rahman barostat [42,43], with a time constant of 5.0 ps, were used. At least 10,000 energy frames were generally recorded, while trajectories were recorded at least 1000 times for NPT simulations and at least 5000 times for NVT simulation. We note that Basconi and Shirts have demonstrated that NVT simulations lead to the same values for the dynamic properties of self-diffusion and viscosity as obtained with NVE simulations as long as proper pressure coupling schemes are used [44]. While NVE simulations have been recommended as the final simulation step to obtain dynamic properties [45], the temperature stability of long simulation times can be problematic.

### 2.2. Analysis

Visual inspection of the trajectories and the generation of pictural representations of the simulated molecules were accomplished by using the VMD freeware [46]. For quantitative analysis of the trajectories, many of the available analysis modules of the GROMACS package were used: the module “energy” to obtain densities, *ρ*, and isobaric heat capacities, *C_p_*, from the NPT simulation at times beyond which density converged, and for the NVT simulations, the modules “msd”, “distance”, “gyrate”, and “rdf” were used to obtain mean square displacements, intramolecular distances, radii of gyration, and radial distribution functions (RDFs), respectively. The RDFs evaluated were between oxygen atoms: hydroxy–hydroxy (OH-OH) and hydroxy–ether (OH-OE), where the latter was repeated excluding OH-OE from the same molecule so that the intramolecular OH-OE RDF was obtained via subtraction.

The diffusive regime was identified from the time-dependent mean square displacement and linearly regressed to obtain the box size-dependent self-diffusion coefficients *D*(*L*) from *MSD*(*t*) = 6 *D*(*L*) *t* + *c* [45]. The self-diffusion coefficient at infinite box size $D_{\infty }$ was obtained from the analytic correction by Yeh and Hummer [47] in Equation (2):(2)$$D_{\infty }=D\left(L\right)+\frac{k_{B}T\xi }{6\pi \eta L}$$
which amounted to less than 10%. In Equation (2), *k_B_* is the Boltzmann constant, *T* is the temperature, *ξ* = 2.837298, and *η* is the simulation viscosity.

As illustrated in previous work, the time decomposition method based on Green–Kubo (GK) integral formalism [48] was applied to obtain viscosities, *η*, where improved statistics were achieved through repeated analysis of the NVT simulation trajectory split up into multiple time blocks.

For the detailed hydrogen bonding analysis, own python scripts extensions making use of the freely available package “MDAnalysis” [49] were employed. A variety of hydrogen bond counting criteria exist including several based on energetics or topology [50]. This study does not seek exact absolute values for the hydrogen bonding number but rather seeks to find relative trends across the different alcohol molecules. Therefore, as long as the hydrogen bond counting scheme is applied consistently, it is irrelevant which scheme is used for the scope of this study. The default MDAnalysis hydrogen bonding criteria were adopted, and these were that acceptor–donor-hydrogen triplet angles are 30° or less and that acceptor–donor distances are within 0.30 nm.

Finally, relative standard deviations (RSDs) were determined either in prior work or in this work from statistical analysis to be 0.5%, 10%, 7%, and 15% for *ρ*, *C_p_*, *D*_o_, and *η*, respectively.

## 3. Results and Discussion

### 3.1. Comparison of Simulated and Experimental Properties

A comparison of simulated and experimental values is presented in tabulated form in the Appendix A section for density, viscosity, self-diffusion coefficient, and molar heat isobaric capacities in Appendix A, respectively, and in graphical form in Figure 1. Note that experimental data for 1-hexyloxymethanol are not available to the best of our knowledge. The experimental densities are generally well reproduced by the simulations. The n-octanol densities of 824.4 kg·m^−3^ and 820.0 kg·m^−3^, obtained from simulations at 298 K with OPLS-AA and CHARMM, respectively, are in good agreement with the 827 kg·m^−3^ and 823.9 kg·m^−3^ reported in prior OPLS-AA simulation studies [30,32], as well as the 809.6 kg·m^−3^ and 815.2 kg·m^−3^ reported in prior AMBER simulation studies [21,31]. The largest disagreement between simulated and experimental densities is observed for 2-pentyloxyethanol (n = 2), where the simulations underestimate the densities. In contrast, the best agreement of simulated viscosities with experimental values is observed in Figure 1b for 2-pentyloxyethanol. Simulated viscosities are otherwise mostly larger than the experimental values by up to a factor of 3. The viscosities from the CHARMM force field follow somewhat better the trend of the experimental data in Figure 1b. According to the Stokes–Einstein Equation,(3)$$D=\frac{k_{B}T}{\xi \pi \eta r}$$
where *ξ* is a dimensionless constant typically ranging between 4 and 6, and *r* is the hydrodynamic radius of the diffusing molecule, the self-diffusion coefficient, *D*, is inversely proportional to the viscosity, *η*. Consequently, the simulated self-diffusion coefficients are generally lower than the experimental values in Figure 1c, especially at the lower temperature of 298 K. The value of 16.6 × 10^−11^ m^2^·s^−1^ (15.3 × 10^−11^ m^2^·s^−1^ without box size correction) obtained from the simulation with the OPLS force field agrees reasonably well with the value of 11 × 10^−11^ m^2^·s^−1^ reported in a prior OPLS-AA force field simulation of n-octanol [30]. Appendix A lists values for (*D η T*^−1^). The values in Appendix A obtained from the simulations are fairly constant, mostly being between 0.3 × 10^14^ and 0.4 × 10^−14^ N·s^−1^·K^−1^, which is in good agreement with the experimental values, which are in the same range. The apparent constancy of these values is expected, according to Equation (3), given that the hydrodynamic radius is essentially identical for the studied alcohols, and *ξ* is unlikely to vary significantly within this group of similar chemicals. The simulated isobaric molar heat capacities, *C_p_*, in Figure 1d are all about a factor of two higher than experimental values. This discrepancy has been explained by Coleman et al., who showed that the inclusion of quantum effects in the analysis corrects this about two-fold overestimate [30]. Possibly, the inclusion of quantum effects would also correct the incorrect temperature trend in the simulated *C_p_* values that the simulated values in Figure 1d are about the same, if not decreased, at 358 K compared to 328 K, while the experimental *C_p_* values [51] are systematically increasing with a temperature increase. Figure 2 shows the Arrhenius activation energies for viscosity and self-diffusion estimated from the simulated data at 298 K and 358 K. While the activation energies obtained from the simulations are all on the order of 10 kJ·mol^−1^ higher compared to the activation energies obtained from experimental values, they do capture the same trend as observed in the experimental data that the activation energies for diffusion are higher than for viscosity. The only clear exception to this trend is observed for the OPLS simulation of 5-ethoxypentanol.

Overall, the simulations reproduce, reasonably well, the experimental values while showing differentiation between the OPLS-AA and the CHARMM force field, where the CHARMM force field appears to better reproduce the trends of the experimental data across the ether alcohols. In our prior study on polyethylene glycol, we could improve the agreement between simulated and experimental viscosities and self-diffusion coefficients by lowering the polarity of the OH group and lowering the potential energy barrier for the OH-C-C-O carbon–carbon bond rotation. We attempted similar modifications to the OPLS-AA force field for 2-pentoxyethanol, but surprisingly, the simulations at 298 K resulted in no changes to the simulated physical properties. Apparently, the presence of at least one additional ether functionality in the molecular structure is important for these force field modifications to notably affect simulation outcomes.

### 3.2. Intramolecular Hydrogen Bonding

Hydrogen bonding can be expected to play a key role in explaining the trends of the physical properties presented in Section 3.1 across the studied alcohols. In this subsection, we begin a detailed analysis of the present hydrogen bonding interactions by inspecting the RDFs of the hydroxy oxygen (OH) and the ether oxygen (OE) within the same molecule (Figure 3) to identify how much intramolecular hydrogen bonding occurs for each ether alcohol. As pointed out in the Introduction, each ether alcohol can maximally engage in only one intramolecular hydrogen bond. When such an intramolecular hydrogen bond is formed, the distance between OH and OE is rather short and is associated with a peak near 0.28 nm in the RDFs in Figure 3. If the ether alcohol does not engage in hydrogen bonding, the present features in the RDFs of Figure 3 reflect the distances between OH and OE along the carbon backbone of the ether alcohol. With an increasing n, i.e., increasing number of carbon atoms between OH and OE, not only does the OH-OE distance increase, but the configuration space for stable configurational isomers increases as well. Thus, multiple distinct features are observable for some of the RDFs in Figure 3. Immediately discernible in Figure 3 is that the temperature increase from 328 K to 358 K does not change the peak positions in the RDFs. Even the peak intensities are changing mostly by only small amounts between 328 K to 358 K.

Beyond these general trends, the specific trends in Figure 3 are as follows: For n = 1 (1-hexyloxymethanol), the RDFs in Figure 3 show a single peak centered at 0.25 nm across all simulation conditions. This single peak is readily explained. 1-hexoxymethanol does not engage in intramolecular hydrogen bonding, as the ring strain would be too high. The ether oxygen is only two bonds away from the hydroxy oxygen, and the single peak is a result of the single angular vibration of the OH-C-OE bond angle. For n = 2 (2-pentyloxyethanol), there are two distinct peaks observable in Figure 3 at 0.30/0.37 nm in CHARMM and 0.28/0.37 nm in OPLS-AA. As we will show later, the ring constraint is still too high for any significant amount of intramolecular hydrogen bonding to occur in 2-pentyloxyethanol. Thus, the peak at 0.28 nm for OPLS-AA is not from intramolecular hydrogen bonding. Instead, the two peaks reflect the two major structural configurations along the OH-CH_2_-CH_2_-OE dihedral of 2-pentyloxyethanol, with the longer distance being associated with the trans configuration and the shorter distance being associated with a configuration closer to cis for OPLS-AA and more gauche for CHARMM. This observed difference between the two force fields is likely a consequence of the different dihedral potential functions as well as the different treatment of the nonbonded 1–4 pair interactions.

Two major peaks are observed for n = 3 (3-butoxypropanol) in Figure 3: near 0.28 nm and at 0.42 nm. The peak near 0.28 nm is from intramolecular hydrogen bonding. It is stronger for the RDFs obtained from the CHARMM simulations compared to the OPLS-AA simulations and slightly increases at higher temperatures for both force fields. This means that intramolecular hydrogen bonding is predicted to be more present for the CHARMM force field, but both force fields predict that intramolecular hydrogen bonding increases at higher temperatures, thereby at least partially offsetting a decrease in intermolecular hydrogen bonding that can be expected at higher temperatures. Both force fields predict the same dominant structural conformation when 3-butoxypropanol is not engaged in intramolecular hydrogen bonding. This confirmation is represented by the peak at 0.42 nm that slightly decreases at 358 K commensurately to the increase of the 0.28 nm feature. All RDFs in Figure 3 for n = 3 also show a minor feature near 0.38 nm that is slightly more distinct for the RDFs from the CHARMM force field.

Intramolecular hydrogen bonding is also indicated by the RDFs in Figure 3 for n = 4 (4-proposxybuthanol) by the distinct peak near 0.28 nm, which is similarly strong for both force fields and increases slightly at the higher temperature of 358 K, as was observed for n = 3. The remaining features in the RDFs for n = 3 are very different between the two force fields. The RDFs from OPLS-AA only show one major feature slightly above 5 nm with a shoulder peak near 0.47 nm. In contrast, the RDFs from CHARMM display four distinct features near 0.43, 0.52, 0.57, and 0.61 nm. Evidently, the CHARMM force field allows for easier dihedral rotations, resulting in a higher diversity of structural configurations.

The intramolecular hydrogen bonding peak near 0.28 nm is much reduced in the RDFs for n = 5 (5-ethoxypentanol) and n = 6 (6-methoxyhexanol), becoming essentially absent in the RDFs at 298 K. Evidently, while three and four carbon atoms separating hydroxyl and ether functional groups provide the flexibility necessary for maintaining ideal hydrogen bond angles, the increased distance and configurational freedom with five or more separating carbon atoms decreases sharply the intramolecular hydrogen bond formation. The RDFs of n = 5 and n = 6 in Figure 3 look similar, with the main feature for n = 5 slightly above 0. 65 nm and, for n = 6, slightly above 0.7 nm, where this feature is very broad. The main difference between CHARMM and OPLS-AA RDFs is the broad feature slightly above 0.55 nm observed in the OPLS-AA RDF for n = 5, which is diminished and broadened out in the corresponding RDF of CHARMM.

Overall, the RDFs in Figure 3 show that intramolecular hydrogen bonding is absent for n = 1 and n = 2, significantly present for n = 3 and n = 4, and only marginally present for n = 5 and 6. Snapshots are shown in Figure 4 typifying the ring-like molecular configurations the molecules form when engaging in intramolecular hydrogen bonding. The snapshots confirm the assignment of the 0.28 nm peak to intramolecular hydrogen bonding.

The formation of the ring-like intramolecular hydrogen bonding structures should lead to decreased average OH-OE as well as OH-C_terminal_ distances. This is, to a certain extent, supported when inspecting these two types of atom-to-atom distances in Figure 5. Specifically, the OH-OE distances deviate the most from an approximate linear trend in Figure 5a,b for n = 3 for CHARMM, as well as n = 4 for both CHARMM and OPLS-AA, and in Figure 5c,d the OH-C_terminal_ distances are lowest in value for n = 3 and n = 4. These observations agree with the RDFs in Figure 3 with respect to the intramolecular hydrogen bonding peak at 0.28 nm.

It is very interesting that the OH-C_terminal_ distances in Figure 5c,d are for all alcohols, including n-octanol, consistently smaller at 358 K compared to 298 K. This suggests that the alcohol molecules respond to the increased available energy not only by faster translational motion but also by increased dihedral bond rotations, leading to less elongated molecular structural configurations. This increased dihedral bond rotation would also facilitate intramolecular hydrogen bonding, explaining the slight increase in intramolecular hydrogen bonding observed in the RDFs in Figure 3 for n > 2. A contributing reason why dihedral bond rotations increase could also be the reduced intermolecular hydrogen bonding interactions expected for higher temperatures. Intermolecular hydrogen bonding will be inspected in the next subsection. We are closing this subsection with one final note that the distances shown in Figure 5 are the average value of a wide distribution of distances due to the configurational space along the linear chain of sp^3^-hybridized carbon and oxygen atoms. However, the shown average values are highly repeatable, where the repeatability is within the size of the symbols, thus leading to the consistent lower OH-C_terminal_ distances at 358 K.

### 3.3. Intermolecular Hydrogen Bonding

Intermolecular hydrogen bonds may form either between hydroxy groups or, in the case of the ether alcohols, also between the hydroxy and the ether functionalities. We begin with the inspection of RDFs describing intermolecular distances between hydroxy oxygen atoms, g_inter,OH-OH_, as shown in Figure 6. These RDFs display strong, sharp peaks centered around 0.28 nm, accompanied by a broader peak ranging from approximately 0.45 to 0.55 nm. Qualitative inspection of the trajectories revealed that the sharper peak is indicative of hydrogen bonding between OH groups, while the broader peak represents the average O_[1]_–O_[3]_ distance in an O_[1]_-O_[2]_-O_[3]_ hydrogen bond triplet. Figure 7a shows an example of a hydrogen bond triplet where the hydroxy O_[2]_ accepts a hydrogen bond from hydroxy O_[1]_ and donates a hydrogen to the hydroxy O_[3]_.

Additionally, a peak centered about 0.38 nm is observed in the g_interOH-OH_ in Figure 6 for n = 1 where the ether functionality is separated from the OH group by just one carbon atom. This 0.38 nm peak is associated with the formation of double-hydrogen-bonded dimers, as shown illustratively in Figure 7b, showing a ring-like structure with two OH-OE hydrogen bonds between like residues. The strength of the 0.38 nm peak in Figure 6a,c suggests that this dimer formation is relatively common in CHARMM simulations of n = 1. This was further confirmed by a quantitative search for the average number per frame of such double OH-OE hydrogen-bonded ring structures across all ether alcohols. The results shown in Figure 8 illustrate the dominance of such double-bonded dimers for n = 1, with notably high average counts ranging from 30 to 90 per frame. In contrast, the count drops below 10 for n = 2, and for species with higher values of n, the occurrence of double-hydrogen-bonded dimers converges toward zero, likely due to steric effects, such as steric hindrance and the increasing number of carbons between OH and OE functionalities. As can be seen in Figure 8, the simulations with the CHARMM force field result in a higher number of double-hydrogen-bonded dimers. The circular formation of these double-hydrogen-bonded dimers restricts additional hydrogen bonding interactions contributing to a reduced system density and viscosity with a corresponding increase in self-diffusion, as is observed in Figure 1 for n = 1.

The dominant formation of double OH-OE hydrogen-bonded dimers for n = 1 should result in a corresponding dominant feature in the intermolecular RDF for the hydroxy oxygen-to-ether oxygen correlation, g_inter,OH-OE_. This expected feature is indeed observable in the g_inter,OH-OE_ graphs shown in Figure 9 at 0.29 nm. The 0.29 nm peak represents intermolecular hydrogen bonding between hydroxy and ether groups and is by far more prominent for n = 1 compared to the other ether alcohols. Consistent with the above-made observations for n = 1, the peak at 0.29 nm is stronger for the simulations with the CHARMM force field. The peak at 0.29 nm in the g_inter,OH-OE_ graphs in Figure 9 is observable for all ether alcohols, which shows that intermolecular hydrogen bonding between the hydroxy and ether functionalities occurs for all ether alcohols. The hydroxy–ether intermolecular hydrogen bonding peaks are weakest for n = 3 for CHARMM simulations but weakest for n = 2 for OPLS simulations. Otherwise, the ordering is the same between the two force fields, with n = 4 consistently showing the second lowest hydroxy–ether intermolecular hydrogen bonding peak, while this peak is similar in strength for n = 5 and n = 6.

Aside from the 0.29 nm peak, a broad peak is consistently present for distances between 0.45 and 0.50 nm. The snapshot in Figure 10a illustrates that this broad feature corresponds to the O_[1]_-O_[3]_ distance in a hydrogen bond triplet, where O_[3]_ represents an OE, and O_[1]_/O_[2]_ are OH oxygen atoms, which is a similar arrangement as in Figure 7a for a hydrogen bond triplet of three OH groups. For n = 2, there is another peak near 0.55 nm in the g_inter,OH-OE_ graphs in Figure 9, which is more prominent for the results from simulations with the CHARMM force field. This peak is a result from the double hydroxy–ether hydrogen-bonded dimers that are compared to n = 1, being reduced but still significantly present for n = 2, as discussed above for Figure 6, Figure 7 and Figure 8. A side effect of OH-OH hydrogen bonding between two ether alcohols is that the ether group of one ether alcohol is coming into a more well-defined position relative to the OH of the other ether alcohol. This gives rise to additional peaks in some of the g_inter,OH-OE_ graphs in Figure 9. Specifically, Figure 10b,d show structures that, respectively, explain the broad peak below 6 nm for CHARMM simulations with n = 3, the broad feature for CHARMM simulations with n = 4, and the relatively sharp feature for OPLS-AA simulations with n = 3. Finally, all peaks in the g_inter,OH-OE_ graphs in Figure 9 are reduced in intensity at the higher temperature of 358 K, where the reduction is most notable for the 0.29 nm peak. This observation is readily explained by an overall reduction in intermolecular hydrogen bonding at higher temperatures due to the increased kinetic energy present for the molecules in the system.

### 3.4. Interplay of Intra- and Intermolecular Hydrogen Bonding

Moving on from discussing the structural aspects of the intra- and intermolecular hydrogen-bonded species, this section focuses on the quantitative comparison of the types of hydrogen bonding interactions. Figure 11 shows bar graphs of average number of hydrogen bonds, differentiating between the three major categories of intermolecular OH-OH and OH-OE bonds and intramolecular OH-OE bonds. Many of the trends discussed in Section 3.2 and Section 3.3 about the various RDFs are confirmed in Figure 11. Intramolecular hydrogen bonding is significantly present only for n = 3 and n = 4, where the CHARMM force field predicts more intramolecular hydrogen bonding for n = 3, while the OPLS-AA force field predicts that to be the case for n = 4. The very large presence of intermolecular hydroxy–ether hydrogen bonding for n = 1 is a result of the formation of the double-hydrogen-bonded dimer ring structure. Higher temperatures reduce the intermolecular hydrogen bonding interactions but slightly increase the intramolecular hydrogen bonding interactions. As pointed out in Section 3.1., the largest disagreement between the two force fields in predicting physical properties is observed for n = 2. Figure 11 shows that this difference is likely a result in the different simulation outcomes for hydrogen bonding. While both force fields indicate an absence of intramolecular hydrogen bonding, simulations with the OPLS-AA force field result in about 30% lower numbers of intermolecular hydrogen bonds than simulations with the CHARMM force field. Hydrogen bonding numbers are also lower for n-octanol with the OPLS-AA force field but somewhat higher for n = 1. Overall, the total number of hydrogen bonds fluctuate less for the simulations with the CHARMM force fields, which might explain that the trends seen in Section 3.1 with respect to n in the physical properties from the CHARMM simulations are smoother than those from the OPLS-AA simulation.

The presented analyses have, thus far, provided insights into the structural aspects of hydrogen bonding in octan-1-ol and related ether alcohols, which, in turn, explain the observed trends in the alcohols, in particular with respect to density. To further understand the influence of hydrogen bonding interactions on viscosity and self-diffusion, we attempt to present details on the dynamics of hydrogen bonds. As stated in the Experimental section, while each simulation was conducted with 2 fs time steps, frames were saved at larger time intervals. Fluctuations of making and breaking hydrogen bonds may have, thus, taken place between the saved frames. To nevertheless gain some insight into the stability of the hydrogen bonds, we analyzed the number of instances of unique hydrogen bond pairs over time by creating histograms of their frequencies. Here, each unique pair is tracked over 500 frames, amounting to 5000 ps regardless of whether the bond remains continuously formed. An example of this analysis outcome is shown in Figure 12a for 6-methoxyhexan-1-ol (n = 6) at 298 K with OPLS-AA, where the x-axis (Time Bonded/ps) reflects the amount of time spent bonded within 5000 ps regardless of continuity. These histograms were then converted into complementary cumulative distribution functions (CCDFs), giving the probability *P*(*t*) that a unique hydrogen bond exists at *t* ps in a 5000 ps window:(4)$$P\left(t\right)=1-\sum_{i=t_{0}}^{t}\frac{Count\left(i\right)}{Total\;Count}\;$$
where *Count*(*i*) approaches the *Total Count* as *t* approaches the time by which all hydrogen bonds have become incoherent. An example of *P*(*t*) is shown in Figure 12b. Each obtained *P*(*t*) was fitted to a single-exponential decay model:(5)$$P_{fit}\left(t\right)=Ce^{-\gamma t}\;$$
where *C* is a coefficient, and *γ* represents the decay constant, which provides a measure for the likelihood that two atoms are hydrogen-bonded in a 5000 ps window. The obtained fit parameters are summarized in Appendix A. Figure 13 shows graphs of *γ* as a function of *n* for the three main types of hydrogen bonding.

The trend with respect to n observed in Figure 13a,b for the decay rate constants of intermolecular OH-OH hydrogen bond is very similar to what can be seen for the simulated self-diffusion constants in Figure 1c, especially at 298 K, where self-diffusion is fastest for n = 2 and at similar values for n = 0 and n > 2. This suggests that translational motion is strongly correlated with the intermolecular hydrogen bonding interactions, which intuitively makes sense. Specifically, larger values of *γ* mean a faster decay of the CCDF, which, in turn, indicates faster dynamics with respect to forming and breaking hydrogen bonds. Faster hydrogen bonding dynamics make it easier for the molecule to move to a new position and, hence, leads to faster self-diffusion. For the specific case of n = 1, the *γ*-values are the largest ones for intermolecular OH-OH hydrogen bonding but the smallest ones for intermolecular OH-OE due to the double-hydrogen-bonded dimer formation (Figure 8) that, evidently, is very stable. The net effect of these two intermolecular hydrogen bonding interactions is that the self-diffusion coefficient for n = 1 is between that of n = 0 and n = 2. In agreement with observations made in prior sections of reduced intermolecular hydrogen bonding for the case n = 2, the *γ*-values are also comparably high for this ether alcohol in Figure 13, especially for intermolecular OH-OE hydrogen bonding. Intramolecular hydrogen bonding is indicated in Figure 13c to be very stable for n = 3 and even more so for n = 4 but much less stable for n = 5 and very short lived for n = 6. The temperature increase from 298 K to 358 K increases, systematically, all *γ*-values, indicating a reduced general hydrogen bonding stability, confirming the reduced hydrogen bonding features in the RDFs in Section 3.2 and Section 3.3, as well as the reduced average hydrogen bond numbers in Figure 11. In this regard, the *γ*-values in Figure 13 increase with the temperature most strongly for intermolecular OH-OE hydrogen bonding, followed by intermolecular OH-OH and least strongly for intramolecular hydrogen bonding, except for n = 6. Thus, intermolecular OH-OE hydrogen bonding is more likely to break than intermolecular OH-OH hydrogen bonding, which is in agreement with experimentally observed higher activation energies for n-octanol compared to the ether octanols (Figure 2).

Another potential hydrogen bonding aspect that would affect the mobility of the alcohols is the presence of hydrogen bonding networks, as such networks would increase the average hydrodynamic radius. Thus, we conducted a hydrogen bonding network analysis, where we counted the number of hydrogen bonds each molecule is engaged in regardless of type, thus including intramolecular hydrogen bonding as well in the count. The smallest network number of one would stand for just one hydrogen bond. Table 2 reports the averages per frame for the number of networks, the number of hydrogen bonds per network, and some statistics related to these numbers, including the maximum number of hydrogen bonds encountered in a network. Further included in Table 2 are also the average numbers for double-hydrogen-bonded dimers (Figure 9) and networks with cyclic ring patterns.

At 298 K, simulations show an average of 180–300 networks per frame, with networks containing 1–3 hydrogen bonds in the first three quartiles. At 358 K, network numbers drop to 150–270 per frame, with networks containing 1–2 hydrogen bonds in the first three quartiles. While the two force fields more or less agree on average network sizes, CHARMM generally shows more network formations than OPLS at both temperatures. Ring-like structures are rarely present in the hydrogen-bonded networks, ranging, on average, from 0.392 to 1.482 per frame. These are most frequent for n = 0, suggesting that ether oxygens impede bulky intermolecular hydrogen-bonded network formation, which might be a contributing factor for the increased density in the ether alcohols (n > 0) compared to n-octan-1 (n = 0). Also rare are networks with a large number of hydrogen bonds, but they exist. The fourth quartile of network sizes contains networks with greater than 30 hydrogen bonds in some cases. CHARMM simulations reveal a clean trend of decreasing cluster sizes from n = 0 to n = 1, followed by a general increase from n = 1 to n = 5, with a slight drop from n = 5 to n = 6, which interestingly aligns reasonably well with simulation and experimental viscosity trends (Figure 3b). In contrast, OPLS-AA networks do not exhibit clear trends, which also matches up with the obtained simulated viscosities in Figure 3b. It appears that CHARMM more accurately captures the hydrogen-bonded network behavior. This finding implies that in order to get viscosities right, simulation models must reproduce hydrogen bonding network formations accurately.

Overall, Table 2 clearly shows that large hydrogen bond networks are rare but consistently exist. Despite that they are rather rare, they appear to be correlated with obtaining accurate viscosities from simulations. The overwhelming majority of molecules are involved in just one or two hydrogen bonds. Nevertheless, given that the average number of networks is on the order of 250, hydrogen bonding is rather pervasive. Specifically, with the total number of hydrogen bonds of about 2, this results in an average of about 500 hydrogen bonds present in a system of 1000 molecules, which indeed is in agreement with the average hydrogen bond counts in Figure 11.

## 4. Conclusions

MD simulations were presented to derive a better understanding of the role of intra- and intermolecular hydrogen bonding in determining the physical properties of n-octanol and related ether alcohols. The separation between hydroxy and ether groups within ether alcohols significantly influences the hydrogen bonding interactions and, thus, the physical properties, including density, viscosity, and self-diffusion coefficients. Important general trends are discernible across both force fields. Intramolecular hydrogen bonding is absent in 1-hexyloxymethanol and 2-pentyloxyethanol (n = 1 and 2), is strongest in 3-butoxypropanol and 4-propoxybutanol (n = 3 and 4), and only marginally present for 5-ethoxypentanol and 6-methoxyhexanol (n = 5 and 6). Intramolecular hydrogen bonding reduces the packing efficiency, which explains the lower densities experimentally observed for 3-butoxypropanol and 4-propoxybutanol compared to the other ether alcohols. The stability of the intermolecular hydrogen bonds correlates with the rate of self-diffusion. In this respect, intermolecular hydrogen bonding is stronger between hydroxy groups than between hydroxy and ether groups, which explains the higher activation energy for translational motion for n-octanol compared to the ether alcohols. Interestingly, the most stable hydrogen bonds appear to be the intramolecular hydrogen bonds formed by 3-butoxypropanol and 4-propoxybutanol. 1-hexyloxymethanol forms a very specific intermolecular double hydroxy–ether hydrogen-bonded dimer of elevated stability. A similar double-hydrogen-bonded dimer is also present for 2-pentyloxyethanol but much less frequent. Other intermolecular hydrogen-bonded structures identified are generally present in both force fields as well and may only vary in the frequency of their occurrence. Consequently, the RDFs inspected in this study generally show the same features at similar or identical distances that only vary in their intensity. Hydrogen bonding networks involving up to about 30 hydrogen bonds, some of which include ring patterns, are present but only at low rates of occurrence. Nevertheless, there appears to be a correlation between the formation of these hydrogen-bonded networks and the dynamic viscosity. An increased temperature reduces intermolecular OH-OH and OH-OE hydrogen bonding but slightly increases intramolecular hydrogen bonding. While intramolecular hydrogen bond formation lowers the average end-to-end distance, a reduction in end-to-end distances at higher temperature has been observed for all studied alcohols, possibly due to facilitated dihedral bond rotations. The main difference between OPLS-AA and CHARMM is their quantitative prediction of the present hydrogen bonding speciation. For example, OPLS-AA predicts significantly less intramolecular hydrogen bonding for 3-butoxypropanol than the CHARMM force field. These differences are likely due to the stiffer dihedral potentials in OPLS-AA compared to the CHARMM force field. The CHARMM simulations more accurately reproduce experimental trends in the physical properties, although, as just summarized, both force fields qualitatively agree on key molecular behaviors.

The obtained insights on hydrogen bonding in the studied ether alcohols are important for understanding the hydrogen bonding behavior in PEG. Of special note is the finding that the optimal intramolecular hydrogen bonding occurs when hydroxy and ether groups are separated by three or four carbons (n = 3, 4). However, in PEG structures, these positions are occupied by carbon atoms. Both force fields show that shorter separations (e.g., n = 2) are unfavorable for intramolecular hydrogen bonding, while longer separations (e.g., n = 5, as in triethylene glycol) may permit mildly stable intramolecular interactions. Possibly, the presence of multiple ether functionalities is important to understand why triethylene and tetraethylene glycol have a high propensity for intramolecular hydrogen bonding forming bicyclic systems. Thus, additional future studies investigating ether alcohols with a higher number of ether moieties may shed further light on this matter.

## Figures and Tables

**Figure 1 molecules-30-02456-f001:**
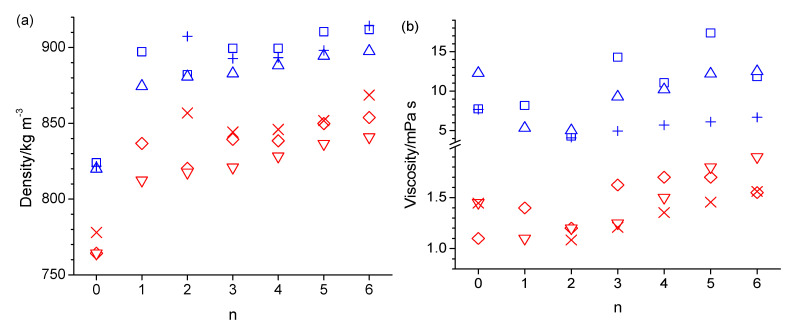
Density (**a**),viscosity (**b**), self-diffusion coefficient (**c**), and isobaric molar heat capacity (**d**) as a function of the integer number of carbon atoms between the hydroxy and ether groups (n) in n-octanol-related ether alcohols, where n = 0 refers to n-octanol, simulated with the OPLS-AA force field at 298 K (blue square) and 358 K (red diamond), with the CHARMM force field at 298 K (blue up-triangle) and 358 K (red down-triangle), and measured experimentally [1,51] at 298 K (blue plus) and 358 K (red cross).

**Figure 2 molecules-30-02456-f002:**
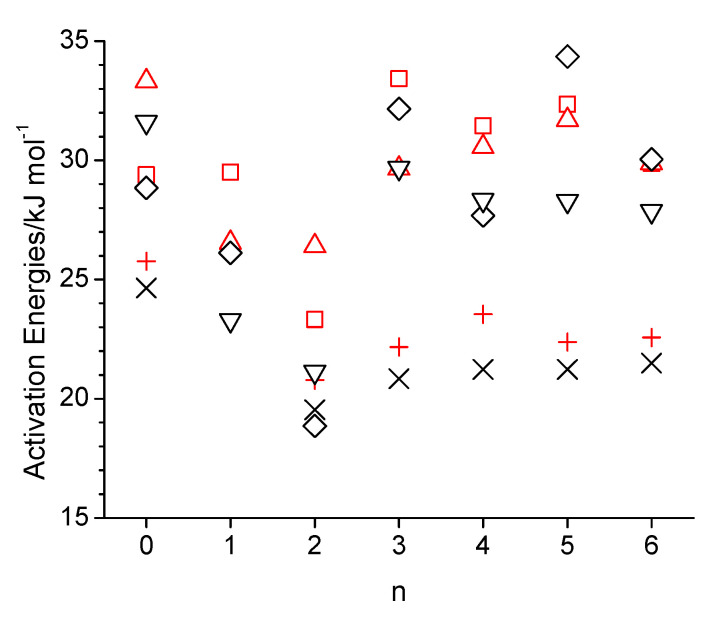
Arrhenius activation energies as a function of the integer number of carbon atoms between the hydroxy and ether groups (n) in n-octanol-related ether alcohols, where n = 0 refers to n-octanol for viscosity (black-colored cross, diamond and down triangle for the experiment [1], simulation with the OPLS-AA force field, and simulation with the CHARMM force field) and self-diffusion (red-colored plus, square, and upside triangle for the experiment [1], simulation with the OPLS-AA force field, and simulation with the CHARMM force field).

**Figure 3 molecules-30-02456-f003:**
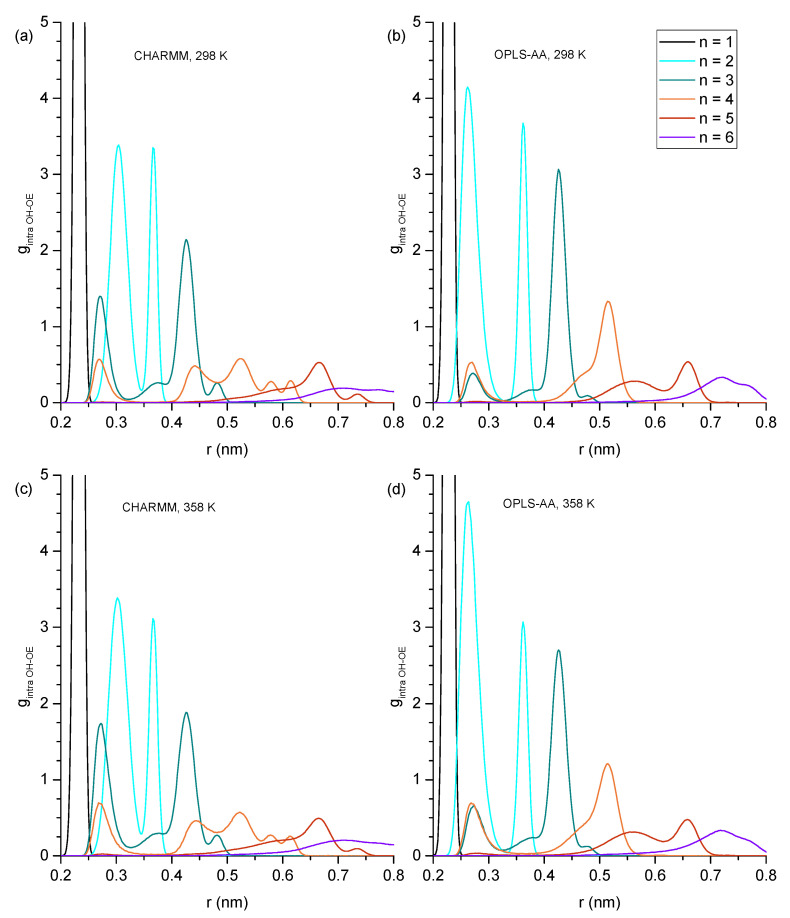
Intramolecular hydroxy–ether radial distribution functions showing probability (g) as a function of radial distance (r) in nm with (**a**) CHARMM, 298 K, (**b**) OPLS, 298 K, (**c**) CHARMM, 358 K, and (**d**) OPLS, 358 K. Values of g greater than 5.0 have been removed for easier inspection of details.

**Figure 4 molecules-30-02456-f004:**
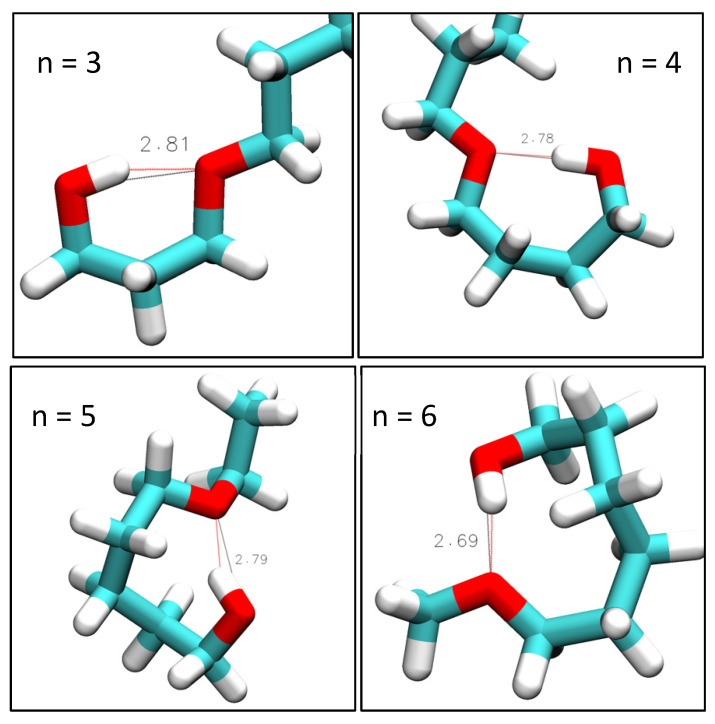
VMD-generated images of intramolecular hydroxy–ether bonds in simulations of n = 2 at 298 K with OPLS, n = 3 and 4 at 298 K with CHARMM, and *n* ≥ 5 at 358 K with CHARMM. Red lines show hydrogen bonding. Black lines show measured hydroxy–ether distance. Indicated distances are in units of Å.

**Figure 5 molecules-30-02456-f005:**
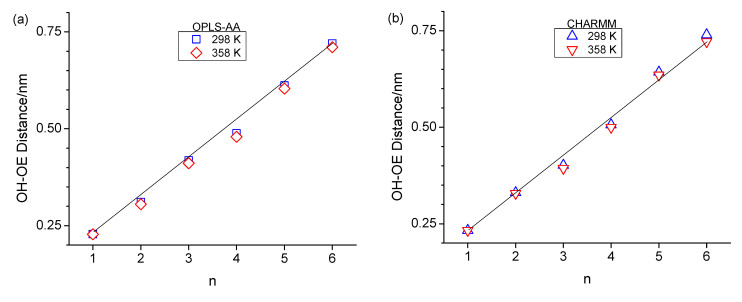
Average intramolecular hydroxy oxygen-to-ether distances (**a**,**b**) as well as average intramolecular hydroxy oxygen-to-terminal carbon distances (**c**,**d**) for OPLS at 298 K (blue squares) and 358 K (red diamonds) and CHARM at 298 K (blue up-triangle) and 358 K (red down triangle) as a function of *n*. The linear line in (**a**,**b**) serves as a guide for the eye.

**Figure 6 molecules-30-02456-f006:**
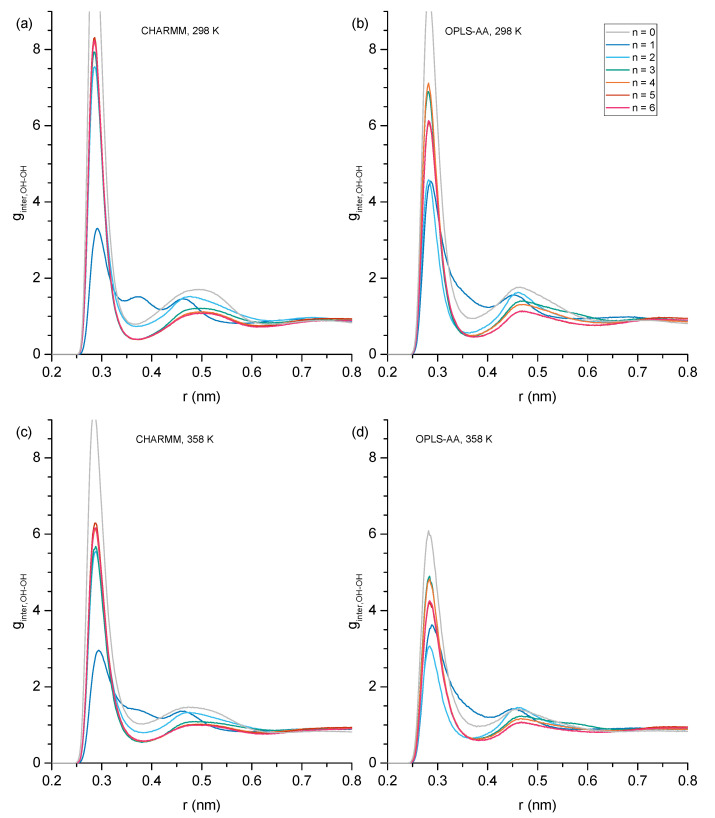
Intermolecular hydroxy–hydroxy radial distribution functions showing probability *g(r)* as a function of radial distance (r) in nm with (**a**) CHARMM, 298 K, (**b**) OPLS, 298 K, (**c**) CHARMM, 358 K, and (**d**) OPLS, 358 K. Values of g greater than 9.0 have been cut off for easier inspection of details.

**Figure 7 molecules-30-02456-f007:**
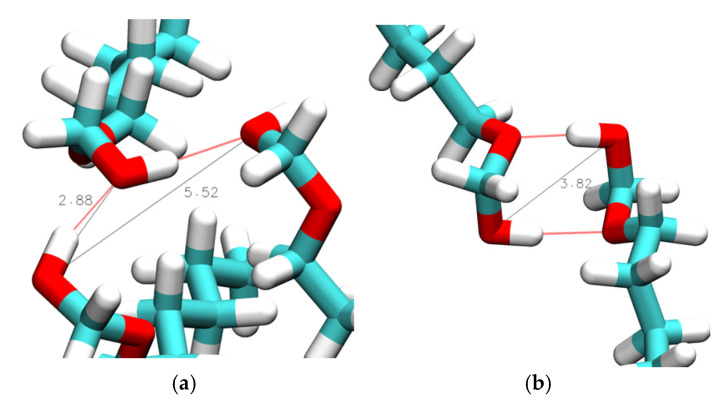
Simulation snapshots of intermolecular hydroxy–hydroxy bond distances in (**a**) 1-hexoxymethan-1-ol (n = 1) and (**b**) a double-hydrogen-bonded dimer in 1-hexoxymethan-1-ol (n = 1) obtained from simulations at 298 K with CHARMM. Red lines show hydrogen bonding, while black lines represent hydroxy–hydroxy distances in units of Å.

**Figure 8 molecules-30-02456-f008:**
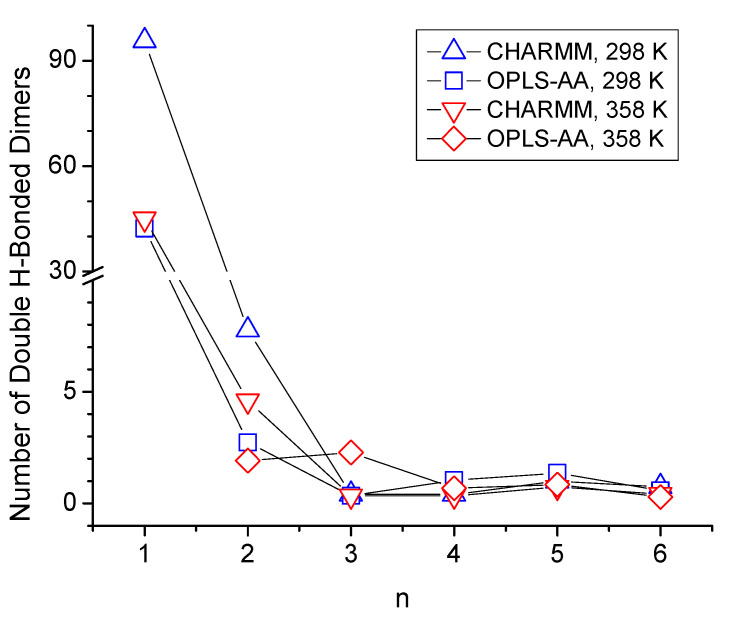
Average number of double-hydrogen-bonded dimers per frame as a function of the number of carbon atoms between hydroxy and ether functionalities, *n*.

**Figure 9 molecules-30-02456-f009:**
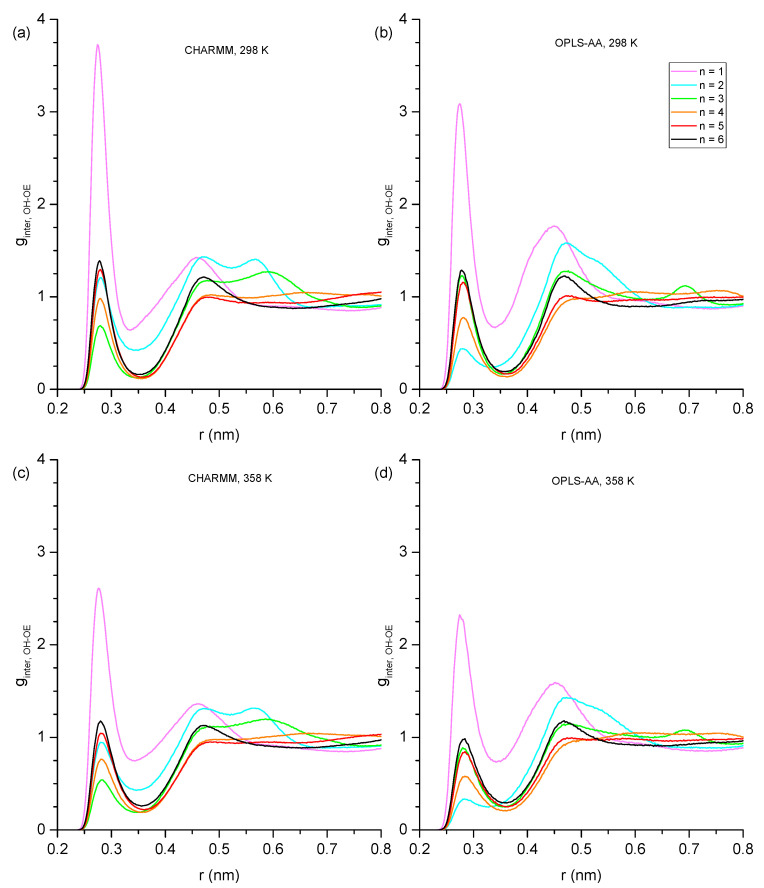
Intermolecular hydroxy–ether radial distribution functions showing probability as a function of radial distance in nm with (**a**) CHARMM, 298 K, (**b**) OPLS, 298 K, (**c**) CHARMM, 358 K, and (**d**) OPLS, 358 K.

**Figure 10 molecules-30-02456-f010:**
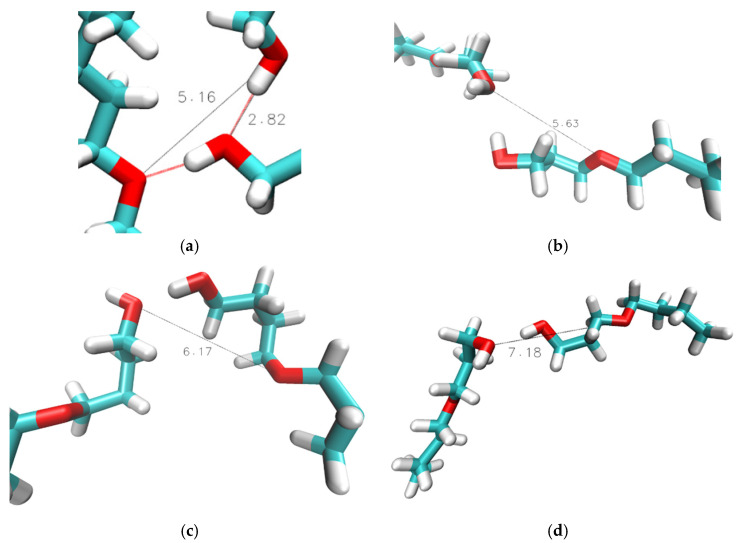
Simulation snapshot of intermolecular hydroxy–ether hydrogen bond distances (**a**) in 2-pentyloxyethanol (n = 2) and (**b**) 3-butoxypropanol (n = 3) and (**c**) 4-propoxybutanol (n = 4) obtained for simulations at 298 K with CHARMM, and in (**d**) 3-butoxypropanol (n = 3) for simulations at 298 K with OPLS-AA. Red lines show hydrogen bonding, while black lines represent hydroxy–hydroxy distances in units of Å.

**Figure 11 molecules-30-02456-f011:**
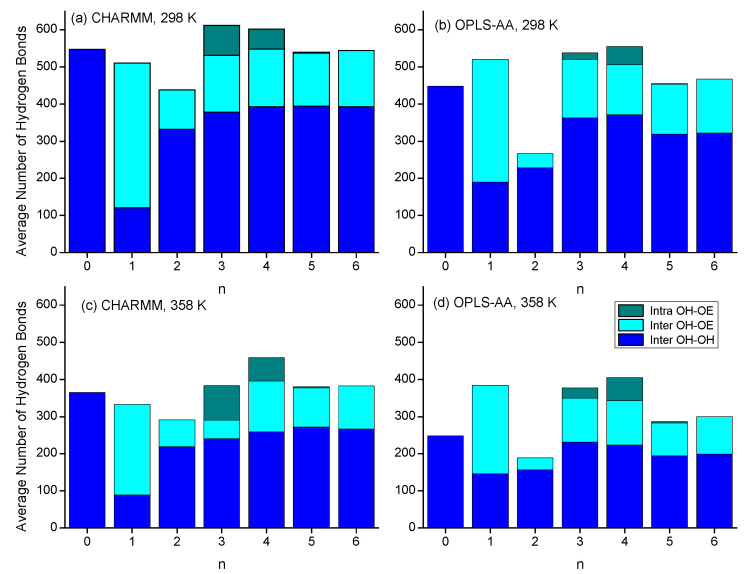
Average number of hydroxy–hydroxy, hydroxy–ether, and intramolecular hydrogen bonds per frame as a function of *n* for simulations with (**a**) CHARMM, 298 K, (**b**) OPLS, 298 K, (**c**) CHARMM, 358 K, and (**d**) OPLS, 358 K.

**Figure 12 molecules-30-02456-f012:**
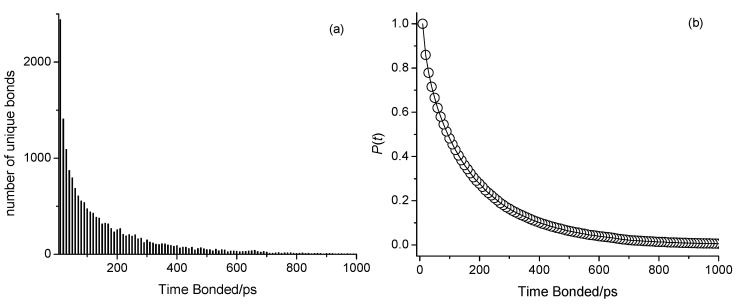
(**a**) Histogram showing the number of unique bonds as a function of time spent bonded in ps, regardless of continuity, within a 5000 ps window for the 298 K CHARMM simulation of n-octanol. (**b**) Complementary cumulative distribution function (CCDF), *P*(*t*), as a function of time spent bonded in ps.

**Figure 13 molecules-30-02456-f013:**
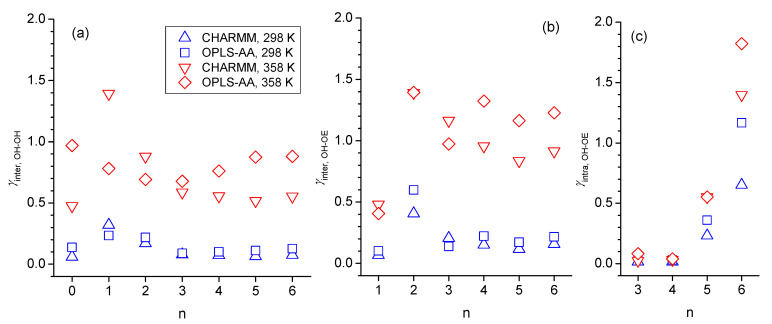
Decay rate constants, *γ*, for fitting the complementary cumulative distribution functions (CCDF) with $P_{fit}\left(t\right)=Ce^{-\gamma t}$ obtained for (**a**) intermolecular OH-OH hydrogen bonding, (**b**) intermolecular OH-OE hydrogen bonding, and (**c**) intramolecular OH-OE hydrogen bonding obtained from simulations with the force fields OPLS-AA at 298 K (blue squares) and 358 K (red diamonds) and CHARM at 298 K (blue up-triangle) and 358 K (red down triangle) as a function of integer n, where n = 0 is for n-octanol that can only engage in intermolecular OH-OH hydrogen bonding. Intramolecular hydrogen bonding was only observed for n > 2.

**Table 1 molecules-30-02456-t001:** Information on simulated chemicals, where *n* is an identifying number for each alcohol.

Chemical Name	CAS	Chemical Structure	n
n-octanol	111-87-5	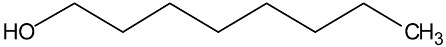	0
1-hexyloxymethanol	44860-30-2	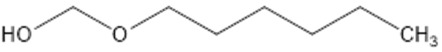	1
2-pentyloxyethanol	6196-58-3	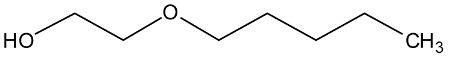	2
3-butoxypropanol	10215-33-5	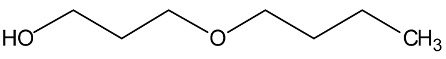	3
4-propoxybutanol	84629-33-4	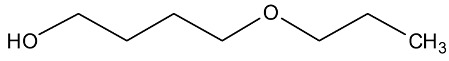	4
5-ethoxypentanol	10215-35-7	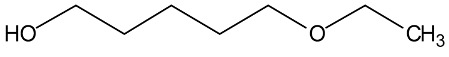	5
6-methoxyhexanol	57021-65-5	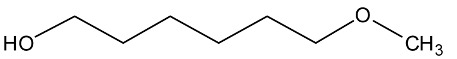	6

**Table 2 molecules-30-02456-t002:** The average number of hydrogen bonding networks per frame, *N*_network_, average number of hydrogen bonds per network, *N*_bonds_, their statistics (Quartile 1, Median, and Quartile 3), the maximum number of hydrogen bonds encountered in a network, *Max*, average number of networks containing double-hydrogen-bonded dimers, *Dimers*, or have a ring pattern, *cyclic*.

n	*N* _network_	*N* _bonds_	Quartile 1	Median	Quartile 3	*Max*	*Dimers*	*Cyclic*
298 K CHARMM								
0	231.1 ± 8.4	2.4 ± 1.9	1	2	3	23	0.0	0.9
1	290.0 ± 8.1	1.8 ± 0.9	1	2	2	11	95.7	0.0
2	240.9 ± 26.6	1.9 ± 1.3	1	1	2	19	7.8	0.1
3	270.2 ± 10.3	2.0 ± 1.5	1	1	2	21	0.4	0.2
4	250.4 ± 9.0	2.2 ± 1.8	1	2	3	27	0.4	0.2
5	214.3 ± 8.5	2.5 ± 2.2	1	2	3	38	1.0	0.3
6	212.4 ± 8.8	2.6 ± 2.3	1	2	3	34	0.7	0.2
298 K OPLS-AA								
0	229.1 ± 7.8	2.0 ± 1.4	1	1	2	17	0.0	1.5
1	252.9 ± 8.4	2.1 ± 1.5	1	2	2	25	42.2	0.0
2	181.7 ± 8.0	1.5 ± 0.9	1	1	2	14	2.7	0.0
3	224.3 ± 8.9	2.3 ± 2.0	1	2	3	28	0.3	0.4
4	248.6 ± 8.6	2.0 ± 1.6	1	1	3	22	1.0	0.6
5	217.7 ± 8.4	2.1 ± 1.7	1	1	3	23	1.4	0.4
6	215.3 ± 8.3	2.2 ± 1.8	1	1	3	23	0.6	0.4
358 K CHARMM								
0	213.7 ± 8.4	1.7 ± 1.3	1	1	2	16	0.0	0.4
1	222.8 ± 9.4	1.5 ± 0.7	1	1	2	11	45.0	0.0
2	191.0 ± 8.4	1.5 ± 0.9	1	1	2	11	4.6	0.0
3	256.7 ± 10.4	1.5 ± 0.9	1	1	2	14	0.3	0.1
4	243.5 ± 9.7	1.6 ± 1.1	1	1	2	16	0.3	0.1
5	209.1 ± 8.2	1.8 ± 1.3	1	1	2	18	0.7	0.1
6	207.9 ± 7.9	1.8 ± 1.4	1	1	2	17	0.4	0.1
358 K OPLS-AA								
0	173.9 ± 9.0	1.4 ± 0.8	1	1	2	11	0.0	0.4
1	226.2 ± 8.3	1.7 ± 1.1	1	1	2	15	24.3	0.0
2	144.4 ± 8.6	1.3 ± 0.7	1	1	1	10	1.9	0.0
3	171.5 ± 8.5	1.4 ± 0.8	1	1	2	11	2.3	0.0
4	229.0 ± 9.5	1.5 ± 0.9	1	1	2	13	0.8	0.1
5	185.8 ± 8.6	1.5 ± 0.9	1	1	2	12	0.9	0.1
6	189.6 ± 8.2	1.6 ± 1.0	1	1	2	14	0.3	0.1

## Data Availability

Data are contained within the article and Appendix A.

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
