# Peer review of "The Interplay of Inter- and Intramolecular Hydrogen Bonding in Ether Alcohols Related to n-Octanol"

_molecules, 2025, doi:10.3390/molecules30112456_

Round 1

Reviewer 1 Report

Comments and Suggestions for Authors

Frankly speaking, I like this study, and I sincerely want it to be published.  Many parts in this manuscript are inspiring, from my point of view.  However, I have to suggest a rejection.

In the methodology part, the authors conducted the simulation in two stages.  Stage 1, a constant NPT ensemble simulation, to general the density, i.e. , the box size; and stage 2, a constant NVT ensemble simulation.  However, the authors extensively talked about time-related properties of the system, such as the diffusion, viscosity, and some microscopic time-correlation properties, which typically need a set of microcanonical ensemble simulations, i.e., constant NVE simulations.  The thermostat and of course barostat can ruin the dynamics of you system, instead of the quantum effect the authors suggest (at least nuclear quantum effect is not the key, because in the simulation, the authors fixed the bond length of hydrogen-contained bonds, which is acceptable to prevent the zero-point energy leakage in classical MD studies).

Author Response

Frankly speaking, I like this study, and I sincerely want it to be published.  Many parts in this manuscript are inspiring, from my point of view.  However, I have to suggest a rejection.

In the methodology part, the authors conducted the simulation in two stages.  Stage 1, a constant NPT ensemble simulation, to general the density, i.e. , the box size; and stage 2, a constant NVT ensemble simulation.  However, the authors extensively talked about time-related properties of the system, such as the diffusion, viscosity, and some microscopic time-correlation properties, which typically need a set of microcanonical ensemble simulations, i.e., constant NVE simulations.  The thermostat and of course barostat can ruin the dynamics of you system, instead of the quantum effect the authors suggest (at least nuclear quantum effect is not the key, because in the simulation, the authors fixed the bond length of hydrogen-contained bonds, which is acceptable to prevent the zero-point energy leakage in classical MD studies).

Author reply: To address this point we added the following sentence in the methods section:

“We note that Basconi and Shirts have demonstrated that NVT simulations lead to the same values for the dynamic properties of self-diffusion and viscosity as obtained with NVE simulations as long as proper pressure coupling schemes are used [43]. While NVE simulations have been recommended as the final simulation step to obtain dynamic properties [44], the temperature stability of long simulation times can be problematic”

Reviewer 2 Report

Comments and Suggestions for Authors

Please see attached review

Author Response

To address the reviewer’s comments, we added one sentence in the introduction:

“A more refined and quantitative hydrogen bonding analysis would require a combination of complementary computational and experimental approaches as illustrated by Pocheć et al for n-octanol [21] of ab-initio metadynamics calculations as proposed by Biswas and Wong [22], which are beyond the scope of this investigation.”

We also added the density result in reference [21] and the density result from another reference mentioned in [21] when comparing simulated results with our obtained densities.

Round 2

Reviewer 1 Report

Comments and Suggestions for Authors

I  did not know that work you cited.  Although it look weird... I did not find some evidence against it.  However, since the authors have cited that research, I would like to suggest publish the paper, instead wasting the authors' time.

However, I would still suggest the authors can use the standard way to get the time-correlation function.

Reviewer 2 Report

Comments and Suggestions for Authors

The authors have addressed my comments; I support publication